# GLoRa: A Benchmark to Evaluate the Ability to Learn Long-Range Dependencies in Graphs

**Dongzhuoran Zhou & Evgeny Kharlamov**
Bosch Center for AI, Germany and University of Oslo, Norway
`{dongzhuoran.zhou,evgeny.kharlamov}@de.bosch.com`

**Egor V. Kostylev**
University of Oslo, Norway
`egork@ifi.uio.no`

## Abstract

Learning on graphs is one of the most active research topics in machine learning (ML). Among the key challenges in this field, effectively learning long-range dependencies in graphs has been particularly difficult. It has been observed that, in practice, the performance of many ML approaches, including various types of graph neural networks (GNNs), degrades significantly when the learning task involves long-range dependencies—that is, when the answer is determined by the presence of a certain path of significant length in the graph. This issue has been attributed to several phenomena, including over-smoothing, over-squashing, and vanishing gradient. A number of solutions have been proposed to mitigate these causes. However, evaluation of these solutions is currently challenging because existing benchmarks do not effectively test systems for their ability to learn tasks based on long-range dependencies in a transparent manner. In this paper, we introduce GLoRa, a synthetic benchmark that allows testing of systems for this ability in a systematic way. We then evaluate state-of-the-art systems using GLoRa and conclude that none of them can confidently claim to learn long-range dependencies well. We also observe that this weak performance cannot be attributed to any of the three causes, highlighting the need for further investigation.

## 1 Introduction

Graph structures are very common in many domains, including molecules in chemistry, social networks on the Web, and road networks in traffic management. Thus, it is not surprising that learning on graphs is among the most active topics in the theory and practice of ML. However, standard ML approaches, such as classic deep neural networks, are difficult to apply to graphs due to their non-trivial structure, and dedicated approaches have been developed for learning on graphs. The most prominent are based on the idea of *Graph Neural Networks* (*GNNs*) (Scarselli et al., 2008), where each layer updates the vector embedding of each node by aggregating the embeddings of the neighbours from the previous layer and combining the result with the previous embedding of the node itself. There are many GNN variants proposed in the literature, from basic GCNs (Kipf & Welling, 2017) and GraphSAGE (Hamilton et al., 2017) to more advanced methods such as Drew-GCN (Gutteridge et al., 2023) with Laplacian encoding (Kreuzer et al., 2021).

Many GNN-based systems have had great success in various tasks in learning on graphs. However, it is also observed that the performance of such systems degrades dramatically when the task involves learning long-range dependencies—that is, when the ground-truth answer for a node in the graph (or, for graph-level tasks, for the full graph) depends on interaction of nodes with large graph distance between them (Alon & Yahav, 2021). However, this property is crucial for tasks in many applications, and essential interactions along several dozen of graph edges are not uncommon. For example, in the Web context, tasks such as social influence prediction (Qiu et al., 2018), social representation learning (Liu et al., 2022), and item recommendation (Fan et al., 2019) rely on the wide spread of information in social network and user-item graphs. In urban data mining, long-range dependencies are evident in

air quality (Iskandaryan et al., 2023) and traffic flow prediction (Li & Zhu, 2021). In internet safety and finance, the challenges of fake news detection (Han et al., 2020) and anti-money laundering (Weber et al., 2018) are critical to detecting such dependencies in the corresponding graphs. Finally, in biochemistry, long-range dependencies emerge in protein-protein interaction (Jha et al., 2022) and drug side effects prediction (Bongini et al., 2022), where the properties are influenced by long chains of atoms in molecule graphs.

Numerous works address the long-range dependency learning issue by trying to identify its causes and suggesting methods to mitigate these causes. Most prominently, the long-range issue was attributed to three phenomena: over-smoothing (Li et al., 2018), which is an observation that in many GNN architectures the embeddings of all nodes converge to the same constant vector as the number of layers increases; over-squashing (Alon & Yahav, 2021), concerning that the information from many nodes that are potentially relevant for the outcome of the target node has to be compressed into a fixed-size vector; and more classical vanishing gradient (Glorot & Bengio, 2010), saying that, as the number of layers increases, the gradient magnitude of the loss usually decreases, thus slowing training down.

Several methods have been proposed to mitigate each of these phenomena. However, as a starting point for this paper, we observe that there are no methods that can with certainty verify whether an ML system is able to learn dependencies of a given length. Indeed, such systems are often evaluated on benchmarks based on real-life datasets (Dwivedi et al., 2023; Morris et al., 2020), whose properties are not known and not controlled enough to give any guarantees. For example, even if an ML system performs relatively well on such a benchmark, we do not know whether the function to learn and the function the system really learned rely on long-range dependencies. There are also a number of dedicated synthetic benchmarks (Di Giovanni et al., 2023; Bodnar et al., 2021; Alon & Yahav, 2021) to evaluate the long-range performance of such systems. Unfortunately, as we will see in more detail later on, these benchmarks are also not suitable for such guarantees, because they either allow simple functions that do not rely on long-range dependencies but perfectly fit the data examples, or, on the contrary, do not have fitting functions expressible in the GNN architectures.

The main contribution of this paper overcomes these shortcomings; in particular, we present a synthetic benchmark (more precisely, an algorithm that generates a benchmark for every dependency length) and a method how to use this benchmark to identify, with certain guarantees, how long dependencies a graph learning system can learn. As our second contribution, we evaluate the state-of-the-art systems using our method, concluding that their dependency length limit is always very modest. As a final contribution, we demonstrate, by means of dedicated experiments, that the degradation of performance with increasing length cannot be attributed to any of the three phenomena. Therefore, further investigation is required to identify the real cause of the long-range dependency learning issue and to develop new methods to mitigate this issue.

The rest of the paper is organised as follows. In Section 2, we first specify the typical graph ML tasks and formally introduce the long-range dependency issue, concentrating on a variant of formalisation that we call *path-aware*. Then we discuss possible causes for this issue identified in the literature and proposed evaluation methods; thus, Sections 2.3 and 2.4 give an overview of the most essential *related work* for this paper. Finally, we argue that these methods are not satisfactory, thus justifying the need for our benchmarks. We present our benchmark, which we call GLoRa (**G**raph **Lo**ng-**Ra**nge dependency benchmark), in Section 3, where we include detailed intuition and formal justifications that GLoRa benchmarks possess the desired formal properties. Then, in Section 4 we report the results of our experiments. The main experiment demonstrates that none of the state-of-the-art systems can learn dependencies beyond the very modest length 11. Secondary experiments demonstrate that this degradation cannot be attributed to any of the three phenomena. We conclude in Section 5. Secondary supporting materials for the claims of the paper are given in the appendices.

## 2 THE CHALLENGE OF LEARNING LONG-RANGE DEPENDENCIES

In this section, we first introduce typical graph ML tasks and graph neural networks as the standard approach to solving these tasks. We then formalise the long-range dependency learning issue, discuss its possible causes, and overview the existing benchmarks for this issue. Finally, we argue that these benchmarks are not satisfactory, thus justifying the need for our benchmark (presented in Section 3).

## 2.1 LEARNING ON GRAPHS

As customary in the context of graph ML, we formalise a *graph* as a triple $(\mathcal{V}, \mathcal{E}, \lambda)$, where $\mathcal{V}$ is a finite set of *nodes*, $\mathcal{E} \subseteq \mathcal{V} \times \mathcal{V}$ is a set of *edges*, which are pairs of nodes, and $\lambda : \mathcal{V} \to \mathbb{R}^n$ is the *node embedding*, which is an assignment of real-valued vectors of some dimension $n \in \mathbb{N}$ to the nodes.

Note here that we assume edges to be directed; undirected graphs, which are sometimes adopted in graph learning, can be represented in this formalisation by assuming that $\mathcal{E}$ is *symmetric*—that is, such that $(v_1, v_2) \in \mathcal{E}$ implies $(v_2, v_1) \in \mathcal{E}$. As we will see later, our benchmark has a version for both the directed and undirected settings, and our experiments show that each considered existing system performs similarly across these versions; however, the directed version has a conceptual advantage, which ensures stronger guarantees (see Section 4.2). Note also that only nodes in our graphs have embeddings; this is just for brevity, and all our ideas apply to graphs with edge embeddings.

The most common ML task on graphs is (supervised) *node classification*, where a system needs to learn, using training examples, a function that assigns a class (called a *label* in this context) from a finite set to each graph node. *Binary* classification is a particular case when there are only two classes, *True* and *False*. Another common task is *node regression*, where the function assigns a numeric value instead of a class. Both classification and regression may be *transductive*, when all relevant nodes (for all training, validation, and testing) belong to the same graph, fully known for training. The more general is the *inductive* setting, where each example may have its own graph (with the same node embedding dimension). Tasks very close in essence to the inductive tasks are *graph classification* and *regression*, where a value is assigned to the entire graph. In our presentation, we concentrate on inductive binary node classification, but all our ideas transfer directly to the other settings.

## 2.2 GRAPH NEURAL NETWORKS

In principle, we can use many ML approaches to graph learning. However, usual methods, such as deep neural networks, have limited applicability in our setting, because graphs may be of arbitrary size (e.g., a model trained on small graphs should be applicable to arbitrarily bigger graphs); moreover, a graph learning model is expected to be agnostic to graph representation—that is, the answer should be *invariant* over reshuffling of nodes and edges in the model's input. Thus, a standard neural architecture for graph learning is *(message-passing) graph neural networks* (*GNNs*); in general form, a GNN $\mathcal{N}$ with $L$ *layers* for classifying nodes with embedding dimension $n$ to classes $C$ is a triple

$$\left( \{\mathsf{Agg}_\ell\}_{\ell \in \{1,\dots,L\}}, \ \{\mathsf{Comb}_\ell\}_{\ell \in \{1,\dots,L\}}, \ \mathsf{Class} \right),$$

where each *aggregation function* $\mathsf{Agg}_\ell$ maps a multiset of vectors of size $n_{\ell-1}$ to such a vector, each *combination function* $\mathsf{Comb}_\ell$ maps two vectors of size $n_{\ell-1}$ to a vector of size $n_\ell$, and *classification function* $\mathsf{Class}$ maps a vector of size $n_L$ to a class in $C$; here, $n_0 = n$ and each $n_\ell$ is the *dimension* of layer $\ell$. The aggregation, combination, and classification functions of a GNN usually depend on learnable parameters of the model, while $L$ and each $n_\ell$, $\ell \in \{1,\dots,L\}$, are fixed in advance or among the hyperparameters. Given an input graph $G = (\mathcal{V}, \mathcal{E}, \lambda)$ with dimension $n$, GNN $\mathcal{N}$ computes, for each $\ell \in \{1,\dots,L\}$, the $\ell$-th *embedding* $\mathbf{v}_\ell$ of each $v \in \mathcal{V}$ as

$$\mathbf{v}_\ell = \mathsf{Comb}_\ell(\mathbf{v}_{\ell-1}, \mathsf{Agg}_\ell(\{\!\{\mathbf{u}_{\ell-1} \mid u \in N(v)\}\!\})), \tag{1}$$

where $\{\!\{\cdot \mid \cdot\}\!\}$ is the multiset constructor, each $\mathbf{v}_0$ is $\lambda(v)$, and each $N(v)$ is the *neighbourhood* $\{u \in \mathcal{V} \mid (u, v) \in \mathcal{E}\}$ of a node $v$ in $G$; then, the final classification of each $v$ is $\mathsf{Class}(\mathbf{v}_L)$.

A GNN example is *graph convolutional networks* (*GCNs*) (Kipf & Welling, 2017) instantiating (1) as

$$\mathbf{v}_\ell = \sigma \left( \sum_{u \in N(v) \cup \{v\}} \frac{\mathbf{W}_\ell \mathbf{u}_{\ell-1}}{\sqrt{|N(v)| \cdot |N(u)|}} \right),$$

where $\sigma$ is a non-linearity *activation* function, such as ReLU or sigmoid, and each $\mathbf{W}_\ell$ is a learnable matrix of parameters (of dimension $n_\ell \times n_{\ell-1}$); it may also be assumed that, for binary classification, the final dimension $n_L$ is 1, and $\mathsf{Class}(v)$ just thresholds $\mathbf{v}_L$.

Many other variants of GNN are suggested in the literature (GINs (Xu et al., 2019), GraphSAGE (Hamilton et al., 2017), GATs (Velickovic et al., 2018), etc.). Moreover, there are a number of proposals to use a known GNN architecture, but apply it to a modification of the input graph $G$ rather

than to $G$ itself; the most prominent such idea is to *rewire $G$* by adding and removing nodes and edges according to some (often quite sophisticated) rule (Gutteridge et al., 2023; Abboud et al., 2022; Alon & Yahav, 2021; Gasteiger et al., 2019b; Topping et al., 2022; Karhadkar et al., 2023). An extreme of this idea is *graph transformers* (Dwivedi & Bresson, 2020; Kreuzer et al., 2021; Rampásek et al., 2022), where an edge is added between each two nodes, and an attention mechanism is employed to learn nontrivial dependencies. As we will see in the next section, several rewiring-based methods have been proposed to mitigate difficulties with learning long-range dependencies. Another idea to deal with long-range dependencies is realised by implicit GNNs (Gu et al., 2020; Liu et al., 2021; Chen et al., 2022), where the number of layers is not predefined but the same transformation is applied many times until (approximate) convergence. We omit the details of these systems here, because in this benchmark paper we largely treat graph learning systems as a black-box.

Intuitively, node classification functions may be characterised by how long the dependencies they rely on are—that is, how far the target node (i.e., the node to classify) may be from the nodes that this node needs to interact with to decide its classification. It has been observed many times in practice that the quality of learning node classification functions using standard graph learning methods (e.g., GCNs) degrades dramatically when the function to learn relies on long dependencies (Gutteridge et al., 2023; Di Giovanni et al., 2023). As we will discuss in Section 2.3, this issue has been attributed to several causes, and a number of methods are proposed to mitigate this issue. However, we first note that, to the best of our knowledge, the long-range dependencies are not yet formalised in the context of graph learning, and only informal explanations have been suggested (Alon & Yahav, 2021; Dwivedi et al., 2022). Since this issue is central to our paper, we next give such a formalisation.

In the following, let, as usual, a *simple path $p$* from a node $v' \in \mathcal{V}$ to a node $v \in \mathcal{V}$ of *length $d$* in a graph $(\mathcal{V}, \mathcal{E}, \lambda)$ be a sequence $v_0, \ldots, v_d$ of distinct nodes in $\mathcal{V}$ such that $v_0 = v'$, $v_d = v$, and $(v_{i-1}, v_i) \in \mathcal{E}$ for every $i$ with $1 \le i \le d$. Let then $G^{p[k,m]}$, for every $k, m$ with $0 < k \le m < d$, be the graph obtained from $G$ by duplicating the part of $p$ between $v_k$ and $v_m$—that is, the graph with nodes $\mathcal{V} \cup \{v'_k, \ldots, v'_m\}$; edges $\mathcal{E} \cup \mathcal{E}'$, where $\mathcal{E}'$ contains an edge $(u^1, u^2)$ for each $(v^1, v^2) \in \mathcal{E}$, such that each $u^h$ with $h \in \{1, 2\}$ is $v'_j$, if $v^h$ is $v_j$ for some $j$ with $k \le j \le m$, and $v^h$ otherwise; and embedding $\lambda'$ extending $\lambda$ with $\lambda'(v'_j) = \lambda(v_j)$ for each $j$ with $k \le j \le m$.

**Definition 1** *A node classification function $f$ for the node embedding dimension $n$ relies on a (path-aware) dependency of length $d$ if there is a graph $G = (\mathcal{V}, \mathcal{E}, \lambda)$ with dimension $n$, a node $v \in \mathcal{V}$, a simple path $p = v_0, \ldots, v_{d-1}, v$, and vectors $\mathbf{a}_i \ne \lambda(v_i)$ in $\mathbb{R}^n$, for each $i$ with $0 < i < d$, such that*

- *for each $i$ with $0 < i < d$, $f(G, v) \ne f(G_i, v)$ where the graph $G_i = (\mathcal{V}, \mathcal{E}, \lambda^*)$ is the same as $G$ except that $\lambda^*(v_i) = \mathbf{a}_i$ (i.e., in particular, $\lambda^*(v') = \lambda(v')$ for every $v' \ne v_i$);*
- *for each $k, j, m$ with $0 < k \le j \le m < d$, $f(G, v) = f(G_j^{p[k,m]}, v)$ for $G_j^{p[k,m]} = (\mathcal{V}^*, \mathcal{E}^*, \lambda^*)$ the same as $G^{p[k,m]}$ except that $\lambda^*(v'_j) = \mathbf{a}_j$;*
- *for each $i, k, j, m$ with $0 < i < d$ and $0 < k \le j \le m < d$, $f(G, v) \ne f(G_{i,j}^{p[k,m]}, v)$ for $G_{i,j}^{p[k,m]} = (\mathcal{V}^*, \mathcal{E}^*, \lambda^*)$ the same as $G^{p[k,m]}$ except that $\lambda^*(v_i) = \mathbf{a}_i$ and $\lambda^*(v'_j) = \mathbf{a}_j$.*

This definition essentially says that there should be a node, $v$, in a graph with a long enough path, $p$, leading to $v$ that is really important for the classification of $v$: it should be possible to distort the embedding of any node in $p$ to change the classification of $v$; moreover, a combination of $p$ with a duplicate of its part where an embedding of a node is distorted should not lead to the change of classification, while distorting both the path and the duplicate should.

We argue that such requirements for a witnessing path reflect the intuition of what a long-range dependency is. The examples of the real-life tasks in the introduction also essentially rely on these requirements. However, we agree that there may be other, inequivalent, formalisations of this intuition (e.g., one may waive the existence of a path to $v$, asking only for a node with some properties not too close to $v$); we add 'path-aware' to the notion to emphasise our choice.

## 2.3 POSSIBLE CAUSES FOR THE LONG-RANGE DEPENDENCY ISSUE

Since learning long-range dependencies is crucial in graph learning, there is a line of research that has focused on identifying and mitigating the causes of performance degradation with increasing dependency length. In particular, this issue is currently linked to three phenomena: over-smoothing (Li

et al., 2018), over-squashing (Alon & Yahav, 2021), and vanishing gradient (Glorot & Bengio, 2010; Bengio et al., 1994). The graph learning community has developed several methods to mitigate these causes, which we discuss next.

*Over-smoothing* (Li et al., 2018) causes the node embeddings to converge to the same vector as the number of GNN layers increases. This may hamper the effectiveness of many GNNs, especially GCNs, when many layers are needed, for example, to capture long-range dependencies. Methods to mitigate over-smoothing include PairNorm (Zhao & Akoglu, 2020), ResGCN and DenseGCN (Li et al., 2019), JKNet (Xu et al., 2018), DropEdge (Rong et al., 2020), GPRGNN (Chien et al., 2021), DAGNN (Liu et al., 2020), GCNII (Chen et al., 2020), IGNN (Gu et al., 2020), EIGNN (Liu et al., 2021), GIND (Chen et al., 2022), and $G^2$ (Rusch et al., 2023).

*Over-squashing* (Alon & Yahav, 2021) refers to the bottleneck of graph learning formalisms, including GNNs, where information from many nodes condense into a fixed-size vector of one node. In dense graphs, the number of paths to a target node may be exponential in the number of the GNN's layers. Therefore, over-squashing can limit learning long-range dependencies in such graphs. Methods to alleviate over-squashing include several variants of DRew (Gutteridge et al., 2023), SP-GCN (Abboud et al., 2022), FOSR (Karhadkar et al., 2023), SRDF (Topping et al., 2022), FA-GCN (Alon & Yahav, 2021), DIGL (Gasteiger et al., 2019b), Mixhop-GCN (Abu-El-Haija et al., 2019), and graph transformers, such as SAN (Kreuzer et al., 2021) and GraphGPS (Rampásek et al., 2022).

*Vanishing gradient* (Glorot & Bengio, 2010) is a common phenomenon in deep neural networks, including GNNs. As the number of layers increases, the gradient of the loss often decreases, slowing or even stopping the training. Well-known techniques to mitigate this in usual neural networks, such as BatchNorm (Ioffe & Szegedy, 2015), ResNet (He et al., 2015), and LSTM (Graves, 2012), have been adapted for GNNs in systems $G^2$ (Rusch et al., 2023), Gradient-Guided Dynamic Rewiring (Jaiswal et al., 2022), and Residual GANs (Lukovnikov & Fischer, 2021).

## 2.4 EXISTING BENCHMARKS AND THEIR LIMITATIONS

Evaluation of the abilities of GNN-based methods to address the long-range dependency issue (e.g., by mitigating the three phenomena) usually employs benchmarks based on real-world and synthetic datasets. We next discuss these benchmarks and argue that they are ill-suited for this purpose.

Many real-world benchmarks, such as Cora and Citeseer (Yang et al., 2016), Cornell and Texas (Pei et al., 2020), and TUDataset (Morris et al., 2020) were not initially designed for long-range dependency studies. Others, such as OGB (Hu et al., 2020) and LRGB (Dwivedi et al., 2022), were introduced specifically to assess the ability to mitigate the three phenomena. However, many of them have already been argued to lack significantly long-range dependencies (Dwivedi et al., 2022; Tönshoff et al., 2023). Moreover, we observe that real-life benchmarks do not suit well for the purpose in general, because we do not have enough control over their functions to learn. In particular, we have no means to ensure that a function that perfectly fits all the examples actually relies on a dependency of a particular length; in fact, we cannot even ensure that all the labels are justified by some properties of the inputs rather than being just noise. Thus, such benchmarks are only useful to conclude that one system is better than another on this particular benchmark, but not to claim with confidence that a specific approach can learn a dependency of a specific length. So, to give such guarantees, we have to employ synthetic benchmarks.

Several synthetic benchmarks designed for long-range dependency evaluation have also been proposed. They include Tree-Neighbours (Alon & Yahav, 2021), h-Proximity (Abboud et al., 2022), Graph Transfer and Ring Transfer (Di Giovanni et al., 2023; Gutteridge et al., 2023), Synthetic Chains (Gu et al., 2020), Color-Connectivity (Rampásek & Wolf, 2021), Conditional Recall, and Tree Max (Lukovnikov & Fischer, 2021). The common principle in their design is that they label nodes in rather simple graphs using a known function that relies on dependency of a specific length. However, we argue that this is not enough: we also need to ensure that *there is no other classification function that fits all the generated examples, but does not rely on long-enough dependency*; moreover, we also need to guarantee that *there exists such a long-range function that is in principle expressible in the tested graph learning architecture*—that is, that there exists an instance of the architecture (e.g., GCN) that realises, in a uniform way, this function (see Barceló et al. (2020) and Grohe (2023) for the details of such expressibility and related notions (Xu et al., 2019; Morris et al., 2019; Grohe, 2023)). Unfortunately, none of the synthetic benchmarks mentioned above satisfy these two properties. For

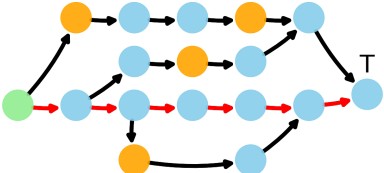 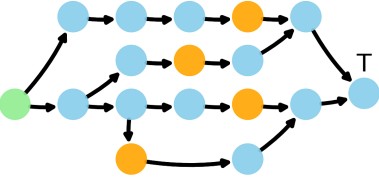

Figure 1: Positive (left) and negative (right) GLoRa examples for $d = 6$: green (i.e., source), blue (normal), and orange (hole) nodes have embeddings of form $[1, -]$, $[-, 1]$, and $[-, 0]$, respectively; T marks the target node; the red-arrows chain is the long-range dependency path

example, in the Synthetic Chains benchmark (Gu et al., 2020), the graph in each example is a directed chain, where the tail node is labelled *True* or *False* depending on whether the start node has 1 or 0 as the first element of its initial embedding vector. These examples are indeed generated by means of a function that relies on a long dependency (in particular, of the length of the longest chain). However, it is possible to perfectly solve this benchmark by a (learned) function that ignores the distance and path information, and only checks whether there somewhere in the graph there is a node without incoming edges with 1 or 0 as the first element. More detailed justification for this and other synthetic benchmarks are in Appendix D. The main contribution of this paper is benchmark generator GLoRa, which produces benchmarks satisfying the two crucial properties.

## 3 BENCHMARK GENERATOR GLoRA

In this section, we present the main contribution of this paper: an algorithm GLoRa that generates benchmarks (i.e., sets of training, possibly validation, and test examples), which can be systematically used to check the ability of a system to learn functions relying on dependencies of a specified length. In particular, our benchmark generated for length $d$ ensures the following properties.

(P1) All node classification functions fitting the training examples rely on dependency of length $d$ with arbitrarily high precision and probability.

(P2) There exists such a function that is expressible by all GNN-based graph learning approaches we are aware of (including GCNs).

(P3) The benchmark is fair—that is, all examples come from the same distribution.

Before we start with intuition, we note that our benchmarks have two versions: one with directed and one with undirected graphs. However, in the following presentation we concentrate on the directed version, while the generalisation to undirected graphs is straightforward.

### 3.1 INTUITION

Our benchmark for the dependency of length $d$ uses graphs with dimension 2 and relies on the following intended binary classification function, which clearly relies on the dependency of length $d$.

(F) The target node is classified as *True* if and only if there is a path of length up to $d$ from a node with embedding of the form $[1, -]$ (which we call the *source* node) to the target node such that all nodes along the path, except the source, have embeddings $[-, 1]$; here, '$-$' represents any number different from 1.

A naive benchmark for this function would use the following examples: a positive example will have a chain of length $d$ or slightly less as the graph, starting from a source node with embedding $[1, -]$, ending at the target node, and such that all nodes along the chain except the source have embedding $[-, 1]$; in turn, each negative example is the same, except that one intermediate node has $[-, 0]$ instead of $[-, 1]$ (in what follows, we call a node with $[-, 0]$ embedding a *hole*).

This approach, however, would not lead to a benchmark that satisfies property (P1). Indeed, it would be possible to perfectly fit all these examples by learning a much simpler function, which does not depend on any significantly long-range dependency; for instance, one such function classifies the target node as *True* if and only if there is no node anywhere in the graph, except the target node, with embedding $[-, -]$ (i.e., where both elements are not 1). Thus, the model that learned this 'shallow' alternative can perfectly classify all the examples of this benchmark.

---

**Algorithm 1** GLoRa Benchmark Example Generation

---

**Input:** Length $d \geq 3$, label $Ans \in \{True, False\}$
**Output:** Graph-node pair $(G, v)$ labelled $Ans$

1: Let $\mathcal{D} := [-12, -2] \cup [3, 13]$

   /* *Part A: Create the main chain* */
2: Let $G := (\mathcal{V}, \mathcal{E}, \lambda)$ with $k_0 \sim \mathsf{U}(\{\lfloor 2/3 \cdot d \rfloor, \ldots, d\})$,
      $\mathcal{V} := \{v_0^0, \ldots, v_0^{k_0}\}$, $\mathcal{E} := \{(v_0^{i-1}, v_0^i) \mid i \in \{1, \ldots, k_0\}\}$,
      $\lambda(v_0^0) := (1, y_0^0)$ with $y_0^0 \sim \mathsf{U}(\mathcal{D})$, $\lambda(v_0^i) := (x_0^i, 1)$ with $x_0^i \sim \mathsf{U}(\mathcal{D})$ for each $i \in \{1, \ldots, k_0\}$
3: **if** $Ans = False$ **then**
4:    Set $\lambda(v_0^{h_0}) := (x_0^{h_0}, 0)$ with $h_0 \sim \mathsf{U}(\{1, \ldots, k_0 - 1\})$

   /* *Part B: Add alternative chains* */
5: Let $P \sim \mathsf{U}(\{5, \ldots, 10\})$
6: **for** $p = 1$ **to** $P$ **do**
7:    Let $k_p \sim \mathsf{U}(\{\lfloor 2/3 \cdot d \rfloor, \ldots, d\})$ and $m_p, n_p \sim \mathsf{U}(\{1, \ldots, k_p - 1\})$ assuming $m_p \leq n_p$
8:    Let $v' \sim \mathsf{U}(\{v_j^{m_j} \in \mathcal{V} \mid m_j = \lfloor k_j \cdot (m_p - 1)/k_p + 1/2 \rfloor\})$
9:    Let $v'' \sim \mathsf{U}(\{v_j^{n_j} \in \mathcal{V} \mid n_j = \lfloor k_j \cdot (n_p + 1)/k_p + 1/2 \rfloor\})$
10:    Add fresh $v_p^{m_p}, \ldots, v_p^{n_p}$ to $\mathcal{V}$ and $(v', v_p^{m_p}), (v_p^{m_p}, v_p^{m_p+1}), \ldots, (v_p^{n_p-1}, v_p^{n_p}), (v_p^{n_p}, v'')$ to $\mathcal{E}$,
       letting $\lambda(v_p^i) := (x_p^i, 1)$ with $x_p^i \sim \mathsf{U}(\mathcal{D})$ for each $i \in \{m_p, \ldots, n_p\}$
11:    Set $\lambda(v_p^{h_p}) := (x_p^{h_p}, 0)$ with $h_p \sim \mathsf{U}(\{m_p, \ldots, n_p\})$

   /* *Part C: Add additional holes* */
12: Let $R \sim \mathsf{U}(\{2, 3\})$ and $\mathcal{V}' := \mathcal{V} \setminus \{v_0^0, v_0^{k_0}\}$
13: **if** $Ans = True$ **then**
14:    Set $R := R + 1$ and $\mathcal{V}' := \mathcal{V}' \setminus \{v_0^i \mid i \in \{1, \ldots, k_0 - 1\}\}$
15: **for** $r = 1$ **to** $R$ **do**
16:    Set $\lambda(v_j^i) := (x_j^i, 0)$ with $v_j^i \sim \mathsf{U}(\mathcal{V}')$
17: **return** $(G, v_0^{k_0})$

---

To ensure (P1), while preserving (P2), GLoRa adds to the chain-like graphs as in the naive approach alternative paths with holes (i.e., $[-, 0]$ embeddings) from the source to the target; this is illustrated in Figure 1. The embeddings on these extra chains also ensure that the number of holes cannot be used as a distinguishing property between positive and negative examples; same should hold for more complex properties, such as the number of holes on paths between a $[1, -]$-source node and the target node. As we will see formally in Theorem 1, such an approach ensures that any GNN-based function fitting (a sufficient number of) such examples rely on a dependency of length $d$. So, we can use GLoRa for length $d$ to check if a system can learn functions with dependency of length $d$ (e.g., $d = 6$ in Figure 1). Moreover, we can use a sequence of the benchmarks with increasing $d$ to identify the limit of a system—that is, the maximal length for which it can learn the dependencies.

### 3.2 GENERATOR ALGORITHM AND GUARANTEES

The pseudocode that generates a GLoRa example is given in Algorithm 1. Since each benchmark is parameterised by length $d$, this number is one of the inputs of the algorithm. The second input, $Ans$, ranges over classes *True* and *False*—that is, specifies whether the example is positive or negative. It is assumed that GLoRa for a given depth $d$ contains a sufficiently large balanced number of examples generated by this algorithm in its training, validation (possibly), and test sets. Note that the algorithm has a large degree of randomness, and the notation $X \sim \mathsf{U}(S)$ means that $X$ is sampled, uniformly at random and independently of other samplings, from a set $S$. The algorithm consists of three parts.

In *Part A* (lines 2–4), the algorithm generates the main chain, which either has no holes, if the example is positive, or one hole (i.e., a $[-, 0]$-embedded node), otherwise. The chain length is not predetermined, but sampled randomly from (rounded) 2/3 of $d$ to $d$, so that the exact length cannot be used by systems in learning. The main chain starts in *source* node $v_0^0$ with embedding $[1, -]$, while all other nodes have embeddings $[-, 1]$ or $[-, 0]$ (i.e., the hole); these nodes include the target node $v_0^{k_0}$.

In *Part B* (lines 5–11), the algorithm adds several (5 to 10) alternative chains with the ends on the previously generated chains (main or additional). The construction ensures (lines 7–9) that there are no loops and the length of the path from the source to the target corresponding to each additional

chain is within the same bounds as the main chain. Each additional chain has a hole, so that only the path along the main chain determines if the example is positive or negative. Note, however, that a model does not know which nodes belong to the main chain, so it has to explore all of them.

Finally, in *Part C* (lines 12–17), the algorithm adds several holes so that each individual path from the source to the target does not have a predetermined number of holes. It also ensures that positive examples remain positive (i.e., that the main chain has no holes), and that the distribution of the number of holes in the positive and negative examples is the same.

We next argue that GLoRa possesses properties (P1)–(P3). Property (P1) is the most sophisticated and is ensured by the following theorem (proved in Appendix C). We use several new notions. First, we introduce an approximate version of Definition 1: a node classification function $f$ *relying on a dependency of length $d$ with precision $\delta > 0$* is the same as in Definition 1 except that $\lambda^*$ in each item is required to match $\lambda$, $\mathbf{a}_j$, and $\mathbf{a}_i$ (when relevant) only with precision $\delta$ (under the maximum norm), while the difference between each $\mathbf{a}_i$ and $\lambda(v_i)$ should have the gap at least $3\delta$—that is, we require, for example, $||\lambda^*(v_i) - \mathbf{a}_i||_\infty \leq \delta$ instead of $\lambda^*(v_i) = \mathbf{a}_i$ for each $i$ and $||\mathbf{a}_i - \lambda(v_i)||_\infty \geq 3\delta$ instead of $\mathbf{a}_i \neq \lambda(v_i)$. Then, a set of examples for node classification *requires learning dependencies of length $d$ with precision $\delta > 0$* if each function that fits all these examples relies on such a dependency.

**Theorem 1** *Let $d \in \mathbb{N}$ be a length. Then, for every $\delta > 0$ and $\mathcal{P} \in (0, 1)$ there exists a number $K$ such that a set of $K$ examples, half generated by GLoRa$(d, \text{True})$ and half by GLoRa$(d, \text{False})$, requires learning dependencies of length $d$ with precision $\delta$ with probability at least $\mathcal{P}$.*

Moving on to property (P2) we note that the results of Barceló et al. (2020) imply that GCNs can express our intended function (F). We also observe that virtually all other known GNN-based classification approaches are more expressive than GCNs, which implies (P2). Finally, property (P3) is automatic since the same (probabilistic) algorithm is used for generating all examples.

In the passing, we note that our algorithm can be easily adapted to undirected graphs.

## 4 EMPIRICAL STUDIES

In this section, we report our results on the evaluation of existing systems' ability to learn long-range dependencies. We begin with the setup of our main experiment, which is based on GLoRa benchmarks. We then present our results, demonstrating that, although some systems perform better than others, none can effectively learn dependencies of significantly long length. Finally, we show that this rather weak performance cannot be attributed to any of the three phenomena.

### 4.1 MAIN EXPERIMENT SETUP AND RESULTS

If a system for learning binary node classification claims that it can learn dependencies of length $d$, then it should show (nearly) perfect testing accuracy on a benchmark generated using GLoRa example generator for length $d$. So, we can find the length limit of a system by running it on such benchmarks with increasing values of $d$ until the accuracy drops from nearly 1 to values close to 0.5.

For our experiment, we generated 5000 positive and 5000 negative examples using Algorithm 1 for each $d \in \{3, \ldots, 15\}$, and split each half with the ratio 80:10:10 to training, validation, and test sets.

Having these GLoRa benchmarks at hand, we evaluated state-of-the-art systems from four categories:

1. *vanilla GNN-based systems* (i.e., not designed to deal with long-range dependencies): GCN (Kipf & Welling, 2017), GAT (Velickovic et al., 2018), GraphSAGE (Hamilton et al., 2017), GatedGCN (Bresson & Laurent, 2017), GGNN (Li et al., 2016), SGC (Wu et al., 2019), and GIN (Xu et al., 2019);

2. GNN-based systems targeting *over-smoothing*: GCNII (Chen et al., 2020), PairNorm (Zhao & Akoglu, 2020), DropEdge (Rong et al., 2020), GPRGNN (Chien et al., 2021), DAGNN (Liu et al., 2020), APPNP (Gasteiger et al., 2019a), ResGCN and DenseGCN (Li et al., 2019), as well as their variants ResGatedGCN, DenseGatedGCN, ResGAT, and DenseGAT with self-explanatory names, JKNet (Xu et al., 2018), IGNN (Gu et al., 2020), EIGNN (Liu et al., 2021), GIND (Chen et al., 2022), and G$^2$ (Rusch et al., 2023);

3. GNN-based systems targeting *over-squashing* (and *vanishing gradient*): $\nu$Drew-GCN, for $\nu \in \{1, \infty\}$, and 1Drew-GCN+LapPE (Gutteridge et al., 2023), SP-GCN (Abboud et al., 2022), FA-GCN (Alon & Yahav, 2021), DIGL (Gasteiger et al., 2019b), MixHop-GCN (Abu-El-Haija et al., 2019), SDRF (Topping et al., 2022), and FOSR (Karhadkar et al., 2023);

4. *transformer-based* systems (Vaswani et al., 2017): GraphTransformer (Dwivedi & Bresson, 2020), SAN (Kreuzer et al., 2021), and GraphGPS (Rampásek et al., 2022), where the last uses either RWSE or LapPE positional encoding.

We run each system on each benchmark five times, training for a maximum of 300 epochs or until convergence, validating hyperparameters, and computing the average testing accuracy over the five runs. For the GNN-based systems, we set the number of layers as $d+2$, which is always enough to express (F). Further details on the experiment setting are given in Appendix A.

In Figure 2, we present test accuracy results for the best five systems in each of the first three categories; detailed results for the other systems, including the transformer-based ones, which perform badly even for very short $d$, are given in Appendix B. Since our benchmark is noise-free, accuracy 1.0 is a reasonable target. Moreover, since this is a binary classification task, random guessing has accuracy 0.5. So, accuracy below 0.8 is an indication that the model has not effectively learned any function that fits the examples—that is, cannot learn dependencies of the given length.

In the figure, we can observe that most systems experience a performance drop before $d$ reaches 9, showing their inability to learn dependencies of significant length. We also see that some systems can learn longer dependencies a little better; however, even their performance is poor after $d = 11$. Notably, the best performance is achieved by systems that use gated edges—that is, GatedGCN, and its variants DenseGatedGCN and ResGatedGCN—suggesting that this feature may be a starting point for GNNs that can truly learn long-range dependencies.

In summary, while some systems perform better than others, none can reliably claim to be able to learn dependencies much longer than of quite modest length 11. We also note that, as we report in Appendix E, the performance of all the systems in the undirected setting is very similar.

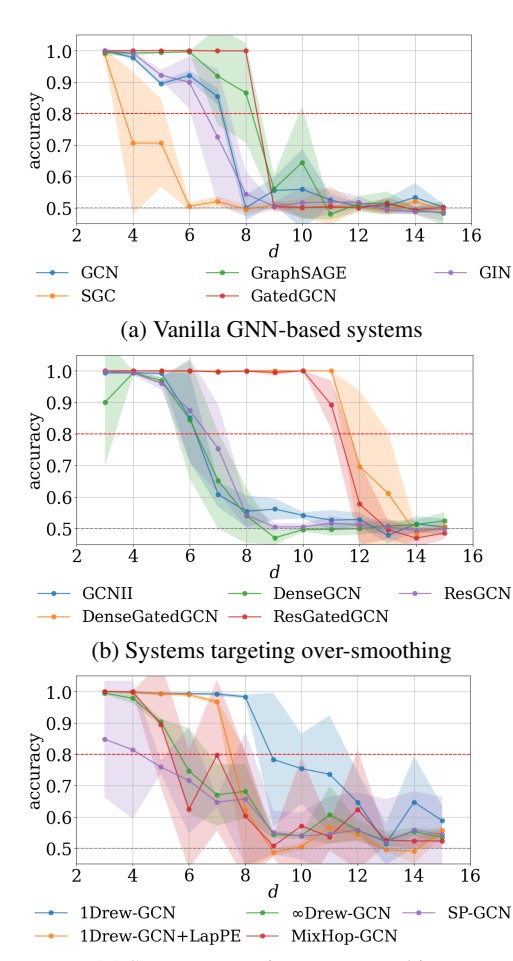

(a) Vanilla GNN-based systems

(b) Systems targeting over-smoothing

(c) Systems targeting over-squashing

Figure 2: Test accuracy of based systems on GLoRa benchmarks for increasing $d$ (lines represent mean accuracy over multiple runs, shaded areas indicate the standard deviation)

## 4.2 OVER-SMOOTHING, OVER-SQUASHING, VANISHING GRADIENT ARE NOT THE REASON

Finally, we argue that, rather surprisingly, none of the three phenomena—over-smoothing, over-squashing, or vanishing gradient—is the reason for the dropping performance in the main experiment. In this section, we focus on the best-performing systems in the three categories (GatedGCN, DenseGatedGCN, and 1Drew-GCN), but the same arguments apply to the other systems.

We first argue that over-smoothing is not a reason for the performance drop in all three systems. To justify our claim, we considered the largest $d$ for which the accuracy is 1 for all systems and

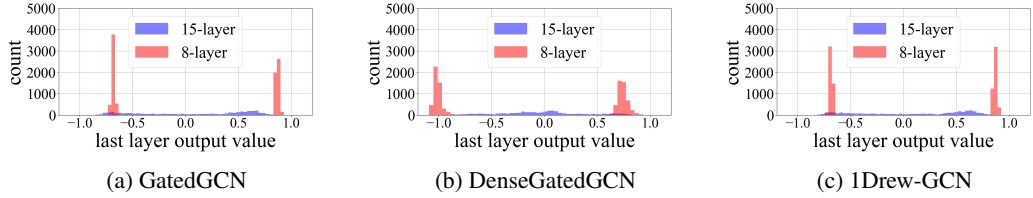

(a) GatedGCN  (b) DenseGatedGCN  (c) 1Drew-GCN

Figure 3: Histograms of the last-layer values for the best-performing systems

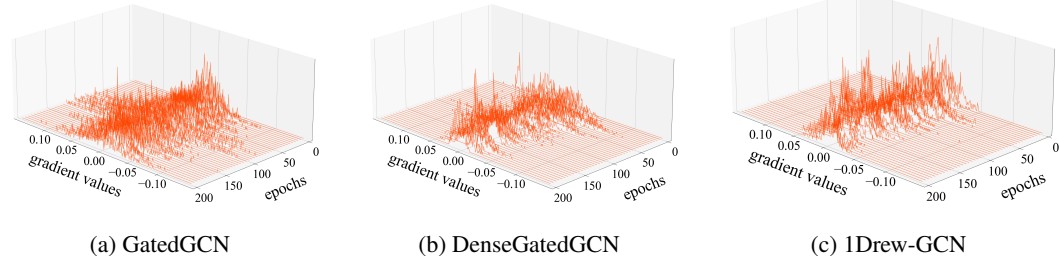

(a) GatedGCN  (b) DenseGatedGCN  (c) 1Drew-GCN

Figure 4: First layer weight gradients across max 300 epochs for best-performing systems

the smallest $d$ for which the accuracy is close to 0.5: 6 and 13. For each case, we extracted the numeric value of the (1-dimensional) last (i.e., 8 and 15) layer embedding of the target node for each of the examples and plotted their histograms in Figure 3. As expected, when accuracy is good, the last-layer embeddings are clearly separated, while when accuracy is poor, they are more mixed together. However, even in the latter case they still do not converge—that is, have rather significant standard deviation. In other words, we do not observe over-smoothing in any of the three cases.

We next argue that over-squashing is also not the reason for degradation for the systems. In fact, this is guaranteed by the construction of GLoRa examples. Indeed, since each example has $P$ additional chains, which is at most 10 (line 5 in Algorithm 1), there is a small (and independent of $d$) bound on the number of paths leading to the target node. Therefore, over-squashing, which concerns the compression of the information from many (exponential number of) paths to a single vector, is not relevant here. Note that directedness of the graphs is crucial for this argument: in undirected graphs, even a single chain to the target node would induce an exponential (in $d$) number of paths leading to this node, and so making such a claim would not be possible.

We finally argue that gradient vanishing is also not the reason for the performance degradation of the three systems. This phenomenon is primarily caused by the fact that the chain rule in backpropagation results in exponentially smaller gradients in previous layers, making training difficult when there are many of them. We plotted, in Figure 4, the distribution of the first layer gradients across multiple epochs for the three systems. As we can see, the gradients remain stable throughout training, with magnitudes well above zero, indicating no vanishing. As we report in Appendix F, the systems have similar behaviour on other layers. This suggests that gradient vanishing is not the reason for their inability to learn dependencies for the corresponding $d$. As a side comment, we note that the standard techniques against gradient vanishing, such as AdamW, weight initialisation, gradient clipping, ReLU activation, and batch normalisation are all employed during training for all the systems.

## 5 CONCLUSION AND FUTURE WORK

The main contribution of this paper is GLoRa, a generator of synthetic benchmarks that allows us to evaluate graph-learning systems in terms of their ability to learn long-range dependencies. Using GLoRa, we have demonstrated that none of the state-of-the-art systems can learn dependencies longer than a very modest length of 11. We have also shown that this degradation cannot be attributed to the over-smoothing, over-squashing, or vanishing gradient. This opens up two directions for future research: to identify the real causes of this degradation and to develop methods to overcome it.

ACKNOWLEDGEMENTS

The work was partially supported by EU Projects Graph Massivizer (GA 101093202), Dome 4.0 (GA 953163), enRichMyData (GA 101070284), and SMARTY (GA 101140087), and the Research Council of Norway through its Centres of Excellence scheme, Integreat—Norwegian Centre for knowledge-driven machine learning, project 332645.

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

# A  MAIN EXPERIMENT DETAILS

In this appendix, we give additional details of the settings of the main experiment in Section 4.1.

## A.1  STATISTICS OF THE BENCHMARK USED IN THE EXPERIMENT

Table 1 presents an excerpt of the benchmark statistics. We created benchmarks for every $d \in \{3, \ldots, 15\}$, but, for readability, give the statistics only for those at intervals of 5—that is, for depths of 5, 10, and 15.

| Length $d$ | Examples | Total Nodes | Average Nodes in a Graph | Mean In-Degree | Total Edges | Average Edges in a Graph | Average $S \to T$ |
|---|---|---|---|---|---|---|---|
| 5 | 10000 | 19334 | 19 | 2 | 259154 | 26 | 5 |
| 10 | 10000 | 336606 | 33 | 2 | 401698 | 40 | 9 |
| 15 | 10000 | 497528 | 50 | 2 | 562624 | 56 | 14 |

Table 1: Statistics of the synthetic benchmark generated by GLoRa with varying length $d$, where each column entry gives the number of the elements in the set of the column name; here 'Average $S \to T$' stands for the average path length from the source node to the target node, while other column names are self-explanatory

## A.2  BENCHMARK LICENSES

Our benchmark generator source code is available at https://github.com/DongzhuoranZhou/GLoRa under the MIT License.

## A.3  SOURCE CODES FOR EVALUATED SYSTEMS

For the *vanilla GNN-based systems* (i.e., not designed to deal with long-range dependencies), GCN, GAT, GraphSAGE, GGNN, SGC, and GIN, we used the implementations included into library PyTorch Geometric (see https://github.com/pyg-team/pytorch_geometric). The implementation of GatedGCN was taken from the repository of GraphGPS (Rampásek et al., 2022).

For the GNN-based systems targeting *over-smoothing*, we used the PyTorch Geometric implementations for GCNII, PairNorm, DropEdge, APPNP, JKNet, ResGCN, and DenseGCN, as well as the variants ResGatedGCN, DenseGatedGCN, ResGAT, and DenseGAT of the last two. The implementations of GPRGNN, DAGNN, IGNN, EIGNN, GIND, and G$^2$ were obtained from the original repositories reported in the corresponding papers.

For the GNN-based systems targeting *over-squashing*, the implementations of $\nu$Drew-GCN with $\nu \in \{1, \infty\}$, SP-GCN, DIGL, and MixHop-GCN were taken from the repository of Drew (Gutteridge et al., 2023), while SDRF and FOSR were taken from https://github.com/hieubkvn123/revisiting-gnn-curvature. Lastly, the FA-GCN architecture was re-implemented in-house.

For all *transformer-based* systems—that is, GraphTransformer, SAN, and GraphGPS with RWSE and LapPE—we used the implementations available in the GraphGPS repository (Rampásek et al., 2022).

The overview of these sources is given in Table 2.

## A.4  COMPUTING ENVIRONMENT

All experiments were conducted in a shared computing cluster environment utilizing various CPU and GPU architectures, including NVidia V100 (16GB/32GB) and NVidia A100 (40GB/80GB). Each system was allocated a resource budget of 1 GPU, 4 CPUs, and up to 32GB of system RAM.

| Category | Baseline | URLs |
|---|---|---|
| vanilla GNNs | GCN, GAT, GraphSAGE GIN, GGNN, SGC | https://github.com/pyg-team/pytorch_geometric |
| | GatedGCN | https://github.com/rampasek/GraphGPS |
| targeting over-smoothing | GCNII, PairNorm, DropEdge ResGCN, APPNP, ResGatedGCN DenseGCN, JKNet, DenseGAT ResGAT, DenseGatedGCN | https://github.com/pyg-team/pytorch_geometric |
| | GPRGNN | https://github.com/jianhao2016/GPRGNN |
| | DAGNN | https://github.com/mengliu1998/DeeperGNN/tree/master |
| | IGNN | https://github.com/sczhou/IGNN |
| | EIGNN | https://github.com/liu-jc/EIGNN |
| | GIND | https://github.com/7qchen/GIND |
| | $G^2$ | https://github.com/tk-rusch/gradientgating |
| targeting over-squashing | $\infty$Drew-GCN, 1Drew-GCN | https://github.com/BenGutteridge/DRew/tree/main |
| | SP-GCN | https://github.com/BenGutteridge/DRew/tree/main |
| | DIGL | https://github.com/BenGutteridge/DRew/tree/main |
| | MixHop-GCN | https://github.com/BenGutteridge/DRew/tree/main |
| | SDRF | https://github.com/hieubkvn123/revisiting-gnn-curvature |
| | FOSR | https://github.com/hieubkvn123/revisiting-gnn-curvature |
| | FA-GCN | https://github.com/DongzhuoranZhou/GLoRa |
| transformer-based | GraphTransformer | https://github.com/rampasek/GraphGPS |
| | SAN | https://github.com/rampasek/GraphGPS |
| | GraphGPS with RWSE and LapPE | https://github.com/rampasek/GraphGPS |

Table 2: URLs for Baseline Model Source Codes

## A.5 HYPERPARAMETER SETTINGS

The essential parameters and hyperparameters are as follows.

- All experiments are trained for 300 epochs or until convergence, with results averaged over 5 runs.

- All experiments use batch sizes of 32.

- All results use the AdamW optimiser with momentum $\beta_1 = 0.9$, base learning rate search space [0.005, 0.01, 0.05], and weight decay space [0, 5e-6, 5e-5, 5e-4].

- All experiments use batch normalisation with $\epsilon = 1e\text{-}5$, momentum=0.9.

- For the GNN-based systems, the number of layers was set to $d + 2$, where $d$ is the depth of the benchmark (which, not surprisingly, was shown to be a best number of layers in all cases in additional experiments).

- For MixHop-GCN, hyperparameter $P$ is a set of integer adjacency powers. The search space for $P$ is [3, 5, 7].

- DIGL and GraphGPS are configured to use GatedGCN as the base model, as it delivers the best performance among MPNNs.

- For DIGL, we employ PPR diffusion with $\alpha$ with search space [0.1, 0.2, 0.5, 0.9] and apply threshold sparsification with average degree $d_{\text{avg}}$ with search space [1, 2, 3, 4, 5].

- GAT head search space is [1, 4, 8].

- JKNet has three options of connection, i.e., concatenation, max, and lstm, which is also taken as a hyperparameter.

- For GPRGNN, we set the random walk path length to $K = 10$, and search for the best PPR within space $\alpha \in [0.1, 0.2, 0.5, 0.9]$.

- For GPS and SAN, layer number search space is $[5, \dots, 14]$.

## B ADDITIONAL RESULTS OF THE MAIN EXPERIMENT

In this appendix, we report the results of the main experiment that are not included in the main body of the paper. In Figure 5, we show the testing accuracy on GLoRa benchmarks with increasing $d$

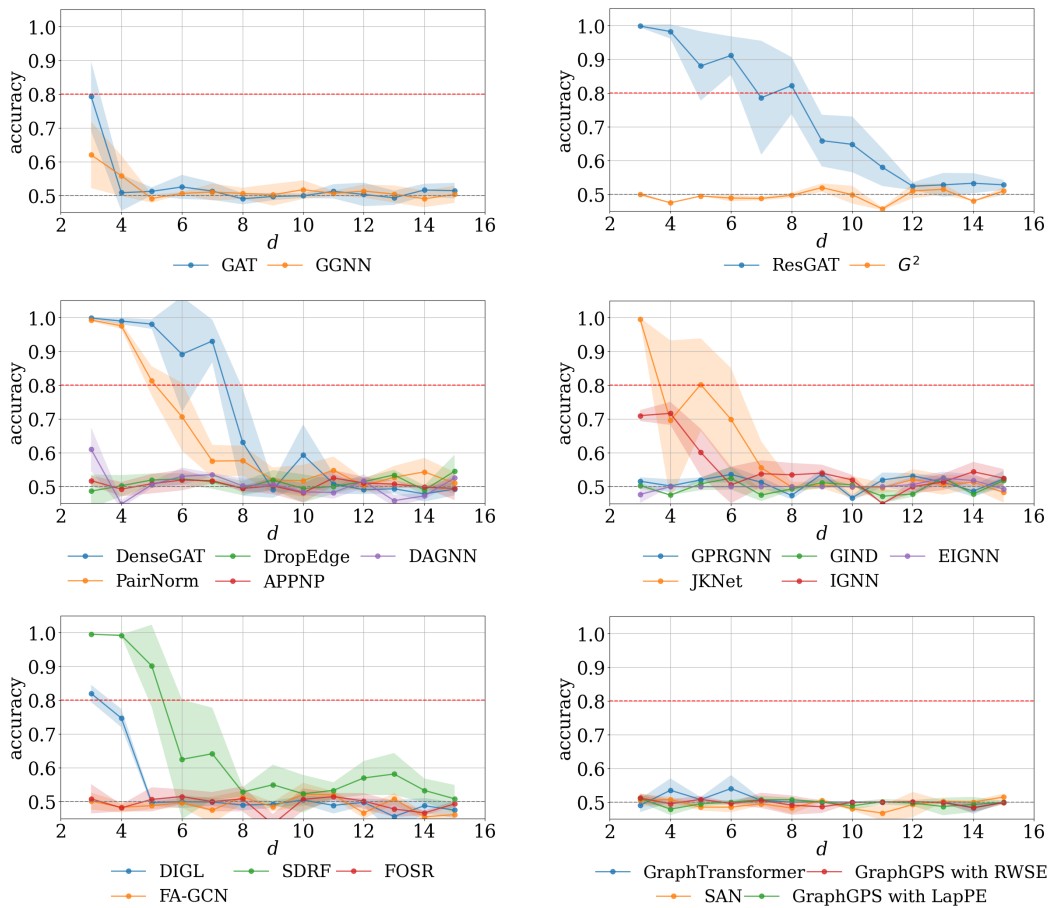

Figure 5: Test accuracy of non-top-5 based systems on GLoRa benchmarks for increasing $d$ (lines represent mean accuracy over multiple runs, shaded areas indicate the standard deviation)

for systems outside the top 5 in vanilla GNNs, GNNs targeting over-smoothing, GNNs targeting over-squashing, and all transformer-based methods. Specifically, for vanilla GNN-based systems they are GAT and GGNN; for the GNNs targeting over-smoothing they are ResGAT, G$^2$, DenseGAT, DropEdge, DAGNN, PairNorm, APPNP, GPRGNN, GIND, EIGNN, JKNet, and IGNN; and for the GNNs targeting over-squashing systems they are DIGL, SDRF, FOSR, and FA-GCN. For transformer-based methods, we consider GraphTransformer, SAN, and GraphGPS.

## C PROOF SKETCH OF THEOREM 1

In this appendix, we give a proof sketch of our theorem.

**Theorem 1** *Let $d \in \mathbb{N}$ be a length. Then, for every $\delta > 0$ and $\mathcal{P} \in (0, 1)$ there exists a number $K$ such that a set of $K$ examples, half generated by GLoRa$(d, \text{True})$ and half by GLoRa$(d, \text{False})$, requires learning dependencies of length $d$ with precision $\delta$ with probability at least $\mathcal{P}$.*

*Proof sketch.* First, observe that, for the given $d$, there is only a finite number of pairs $\mathcal{V}, \mathcal{E}$ of sets of nodes and edges that can be the first two components of a graph generated by GLoRa algorithm with parameter $d$. Moreover, each of these pairs has non-zero probability of being generated. Moreover, $\mathcal{D}$ is bounded and the values are sampled from this set by the agorithm uniformly. So, for the given $\delta$ and $\mathcal{P}$, there is a number $K$ such that $K$ generated examples include, with required probability, an appropriate number of examples that support the claim—that is, such thar the approximated version of Definition 1 holds. Indeed, we only need to ensure that the graph structures of each

positive-negative pair agree with the required structure and between the two examples in the pair, and that their embeddings are within $\delta$ everywhere, except the designated differences. □

## D NON-SUITABILITY OF EXISTING SYNTHETIC BENCHMARKS

In this appendix, we provide a summary of existing synthetic benchmarks, each with its unique graph structures and target functions. Then we explain why each of the existing synthetic benchmarks are not suitable for the purpose of checking whether a system is able to learn long-range dependencies.

Summary of existing synthetic benchmarks is as follows:

- **Synthetic Chains (Gu et al., 2020)**. The graph structure in this benchmark is directed chain. The task is to classify nodes in a directed chain of length $l$. For simplicity, a binary classification is used, with two types of chains. The class of each node is determined by the 1/0 encoding in the first dimension of the starting node's 100-dimensional feature vector. The training set includes randomly sampled nodes from these chains. The target function is that the node is true if and only if the starting node, which has no incoming edges, has a value of 1 in its first dimension.

- **Graph Transfer, Ring Transfer (Di Giovanni et al., 2023; Gutteridge et al., 2023)**. The graph structures in these benchmarks are ring, crossed-ring, and clique-path. Ring graphs are cycles of size $n$. In each ring, there exist pairs of nodes that are placed at a distance of $\lfloor \frac{n}{2} \rfloor$ from each other. One of these node pairs is chosen. One node in the pair has its class information encoded as a one-hot vector (5 classes total) in its embedding, which is used to classify the other node. The target function is that there exists a node with a one-hot encoding node embedding in the graph. The class of the node at a distance of $\lfloor \frac{n}{2} \rfloor$ from this node is the same as the one-hot encoded class. A similar setting applies in crossed-ring and clique-path.

- **Tree-Neighbours (Alon & Yahav, 2021)**. The graph structure in this benchmark is a directed binary tree, where the root and leaves have some additional 1-hop neighbours pointing to them. The target function is that the class of the root node is the class of the leaf node, which has the same number of 1-hop neighbors pointing to it as the root node. Here, the 1-hop neighbours do not include the nodes on the original binary tree.

- **h-Proximity (Abboud et al., 2022)**. The graph structure in the benchmark is defined as follows: (i) consecutive level nodes are pairwise fully connected, (ii) nodes within the same level are pairwise disconnected. Each node is either red, blue, or uncoloured. The target function states that the graph is positive if all red nodes in this graph have at most 2 blue nodes within their $h$-hop neighborhood.

- **Colour-Connectivity (Rampášek & Wolf, 2021)**. The graph structure in the benchmark consists of 2D square grids. Each node is either red or blue. Red nodes are sampled by random walks starting from two random nodes until half the nodes are red; the remaining nodes are colored blue. The target function states that the graph is positive if there is a single connected island of red nodes, and negative if there are two disjoint islands of red nodes.

- **Conditional Recall (Lukovnikov & Fischer, 2021)**: The graph structure in Conditional Recall is a directed chain. The nodes in the chain are encoded as letters (both uppercase and lowercase) and digits. The target function is as follows: (i) if there is a digit in the sequence, the first digit corresponds to the class label of the chain; (ii) otherwise, if there is an uppercase letter, the first uppercase letter is the class label; (iii) if neither digits nor uppercase letters are present, the class is given by the first character in the sequence.

- **Tree-Max (Lukovnikov & Fischer, 2021)**. The graph structure in Tree-Max is a directed binary tree. The nodes in the tree are encoded as digits. The target function is that the label of the node is the largest value of all the descendants of a node and the node itself.

The reason why each of the existing synthetic benchmarks are not suitable for the purpose of checking whether a system is able to learn long-range dependencies are as follows:

- **Synthetic Chains**. Intuitively, to solve this benchmark, it is possible for the learned function to ignore the distance and path information, making global readouts like Virtual Nodes

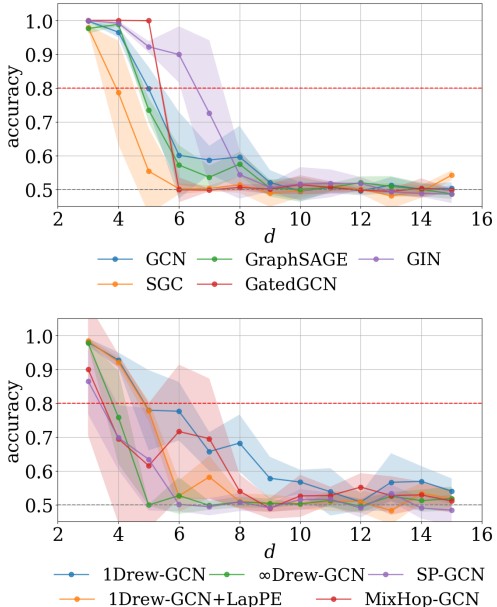
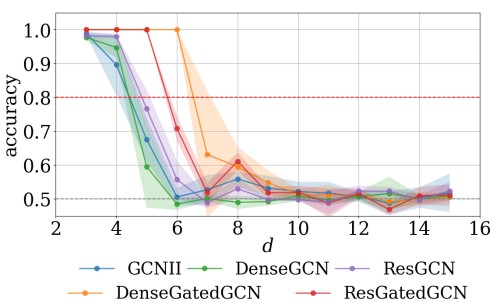

Figure 6: Test accuracy of best five systems in each category of vanilla GNN-based systems, systems targeting over-smoothing and systems targeting over-squashing on undirected GLoRa benchmarks for increasing $d$ (lines represent mean accuracy over multiple runs, shaded areas indicate the standard deviation)

sufficient. In particular, such function does not rely on long-range dependency, because it does not satisfy the requirement that any intermediate node on a justifying path can be dropped.

- **Graph Transfer and Ring Transfer**. This benchmark has similar properties as Synthetic Chains and hence also can be perfectly fitted with a similar function that does not rely on long-range dependency. Moreover, the design of these benchmarks allows only for five different examples (one for each class of the used multi-class classification), which makes the benchmarks of limited use, because these five examples may be memorised by the system during training.

- **Tree-Neighbours**. Same as the previous two cases, this benchmark can be perfectly fitted with a function that does not rely on long-range dependency, which is based on the same principle.

- **h-Proximity**. In this case, the possible function that does not rely on long-range dependency is the one that does not rely on the embeddings of the nodes on the paths between any red node and the target node (thus not satisfying Definition 1).

- **Colour-Connectivity**. On the contrary to the previous cases, this benchmark indeed requires learning function relying on long-range dependency. However, it has an opposite issue: all the GNN-based systems we are aware of cannot express any function that fits a sufficient amount of randomly generated training examples.

- **Conditional Recall, Tree-Max**. The graphs in these benchmarks have limited size, so it is also possible to solve each of them by using a function that does not depend on long-range dependency.

# E   UNDIRECTED GRAPH EXPERIMENT RESULTS

In this section, we present the test results for all the based systems discussed in Section 4 on the undirected version of GLoRa. The results are reported in Figures 6 and 7.

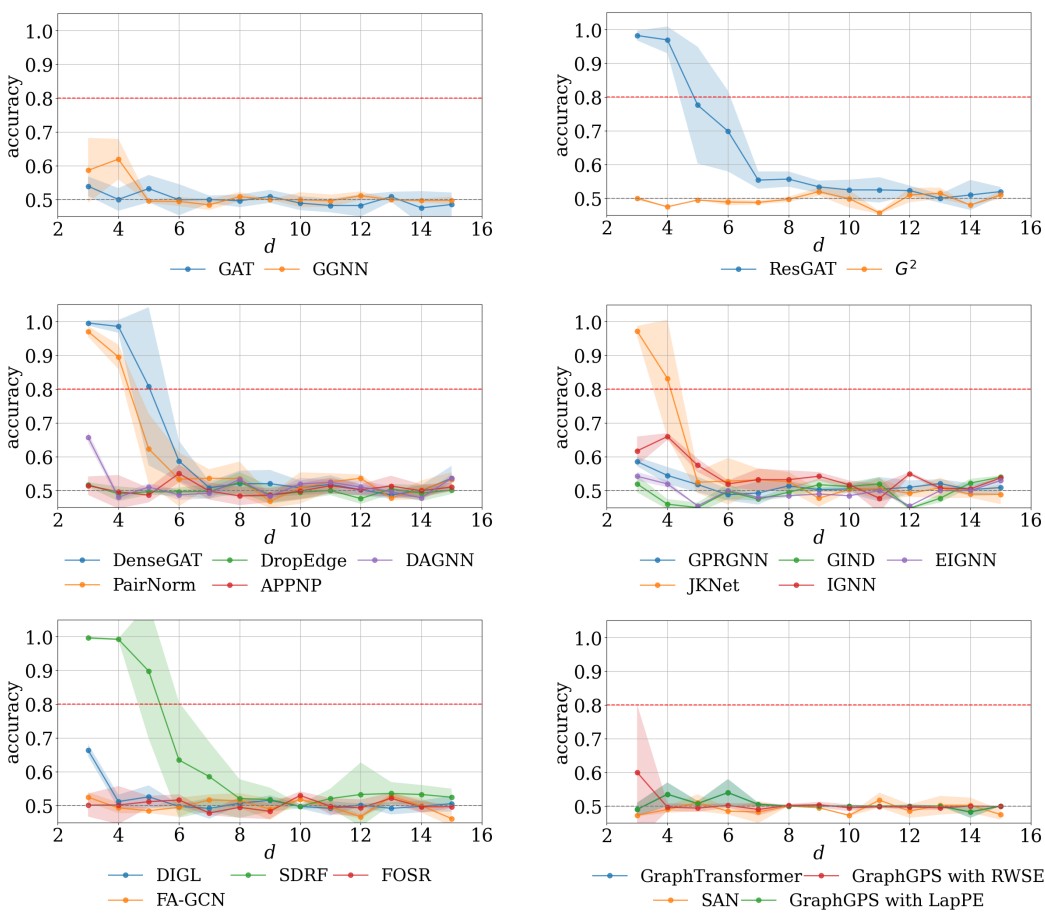

Figure 7: Test accuracy of non-top-5 based systems on undirected GLoRa benchmarks for increasing $d$ (lines represent mean accuracy over multiple runs, shaded areas indicate the standard deviation)

## F   GRADIENT VISUALIZATION OF THE DEEP LAYERS

In this section, we present the gradient distributions of the second and third layers for the best-performing systems. As shown in Figure 8, the gradients in both the second and third layers remain sufficiently large, indicating that the models do not suffer from vanishing gradient problems. This stability of gradients across deeper layers suggests that the degradation in performance is not due to gradient vanishing, but likely other factors.

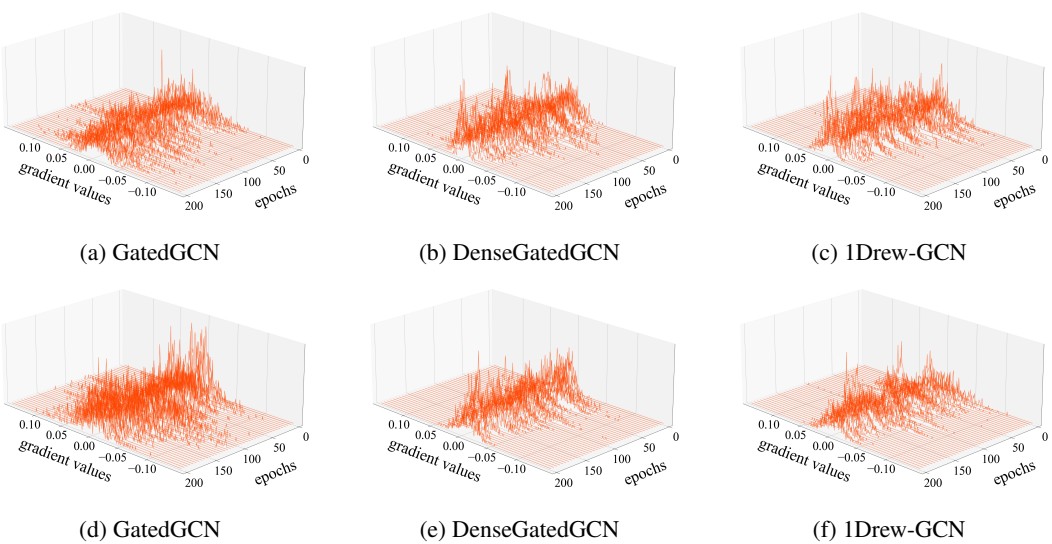

Figure 8: Gradient weights of the second (a, b, c) and third layer (d, e, f) across 300 epochs for the best-performing systems

