# OpenReview forum: "GLoRa: A Benchmark to Evaluate the Ability to Learn Long-Range Dependencies in Graphs"
_ICLR.cc/2025/Conference — ICLR 2025 Poster_

### Official Review · Reviewer_PPjm · 2024-10-30

**Soundness:** 2
**Presentation:** 2
**Contribution:** 3
**Rating:** 8
**Confidence:** 3

**Summary:**

This work introduces a new synthetic benchmark, GLoRa, to measure the ability for graph machine learning methods to learn long-range dependencies. For different depths of dependencies and difficulty levels, GLoRa can make synthetic tasks that do not have simple shortcuts that other long-range benchmarks suffer from. Experiments are conducted across many different GNN architectures from different families. It is argued that oversmoothing, oversquashing, and vanishing gradients are not the issues with learning these long-range dependencies.

**Strengths:**

1. Good criticism of existing benchmarks. Many of them can be solved in a graph independent way, or long-range-dependency independent way.
2. Interesting that over-squashing is not a problem in directed GLoRa by design (number of paths constant and small).
3. Experiments on many types of representative GNNs from different families.
4. In my opinion, good synthetic data experiments were very much needed for this exact question (long-range dependences in graph learning). Doing this well can be quite helpful.

**Weaknesses:**

1. For the definition of path-aware dependencies, isn't it easy to satisfy this for node classification functions on complete graphs, even though the connections are very short. In particular, there is no requirement in this definition for non-existence of paths.
2. Unrigorous proof of Theorem 1
3. 80% in Figure 2 is a bit arbitrary of a threshold, but this isn't a huge issue.
4. Algorithm block is a bit hard to understand, but Figure 1 is clear at least.

**Questions:**

1. Figure 3 does not rule out the case of this type of oversmoothing: at the last layer of a GNN, it may be the case that most nodes in one graph have the same embedding. But, this embedding can be different across different graphs that you run a forward pass on.
2. Why do the Transformers fail, and why do you say this is "not surprising"?
3. Some important details, like how number of layers is chosen, is hidden in Appendix.
4. What is the way forward for making GNNs that solve GLoRa?

---

### Official Review · Reviewer_VTpd · 2024-11-03

**Soundness:** 2
**Presentation:** 3
**Contribution:** 2
**Rating:** 5
**Confidence:** 4

**Summary:**

In this work, the authors propose GLoRa, a benchmark generator for long-range dependency tasks on graphs. The authors overcome the limitations of existing work by precisely stating a definition for long-range dependencies and designing a benchmark that guarantees that models cannot solve the generated tasks unless they respect the long-range dependencies in a given graph. An empirical study on a variety of GNN and transformer baselines concludes that no architecture can perform well on the GLoRA tasks for dependencies longer than depth 10. Further, the authors find that neither over-squashing, over-smoothing or vanishing gradients are the cause for this poor performance.

**Strengths:**

- **S1**: The authors give a clear definition of long-range dependency which is often lacking in prior work.
- **S2**: The authors benchmark a variety of baselines, from simple GNNs to more involved approaches, as well as transformers.
- **S3**: The finding that neither over-smoothing, over-squashing or vanishing gradients are the causes for the poor performance at long range is very interesting and deserves more attention in future research.
- **S4**: The authors have identified a surprisingly simple problem setting that a variety of GNN architectures fail to solve. These findings could lead to interesting follow-up work which aims to understand why the models fail and how these limitation can be overcome.

**Weaknesses:**

- **W1**: The authors argue for the need of a synthetic benchmark where long-range dependencies can be guaranteed. The authors argue that such guarantees are not given in real-world benchmarks. While I generally agree with the fact that in real-world benchmarks there may be shortcuts or simpler functions that avoid long-range dependencies but still correctly predict the labels, I am concerned that the present work proposes a benchmark for a problem they cannot identify in the real-world. In particular, the authors argue in the introduction that long-range dependencies are crucial in many applications (recommendation systems, traffic prediction, fake news detection). However, if long-range dependencies are crucial in these applications I would argue that it would be more sensible to derive benchmarks from real-world data in these domains. Further, if the authors are concerned that in real-world data one cannot verify whether long-range dependencies are truly needed to solve a task, I conclude that the authors also cannot guarantee that their proposed notion of long-range dependencies (Definition 1) is actually useful in real-world applications. Hence, I ask the authors to justify the relevance of long-range dependencies in real-world problems or to argue otherwise how progress on GLoRA contributes to solving real-world problems in graph learning.
- **W2**: The theoretical contributions are formally imprecise and no full proof is given for Theorem 1 (only a proof sketch). First, the authors should clearly state in L385 that Theorem 1 is only supported by a proof sketch. The authors say “proof” in the main paper but “proof sketch” in the appendix. Second, let me expand on what I mean with formally imprecise. The statement of Theorem 1 says “[For every probability $\mathcal{P}$] there exists a number $K$ such that a set $S$ of $K$ samples […] requires learning dependencies of length $d$ with probability at least $\mathcal{P}$.” It is not formally clear what it means for a set of samples to require learning dependencies of some length. I could guess that the authors mean that a model cannot separate the set into positive and negative samples unless the model finds the corresponding path of length $d$. However, the statement in its current form is not formally precise. The authors should either formally state their theorem and prove it, or replace the theorem with an informal argument supported by the proof sketch. If the authors decide to formally state their theorem they should carefully define what they mean with the statement above. It is perfectly fine to give an informal (but intuitive) version of the Theorem in the main text and precisely formalize the theorem statement and proof in the appendix. In this case, I recommend to state the theorem in the main text as "Theorem 1 (informal)" and then write something like "For a precise theorem statement and formal proof, see Appendix ...".

**Questions:**

- **Q1**: The experimental setting is not fully clear to me. In L412 the authors state that they “generated 1000 positive and 1000 negative examples by Algorithm 1 for each $d \in \{3, \dots, 15\}$”. Does that mean that models are trained on $2000 \cdot 0.8 = 1600$ training samples for each depth $d$ or do you construct a joint training set of size $2000 \cdot 0.8 \cdot 13 = 20,800$ training samples and merely evaluate the test accuracy separately for each depth $d$?
- **Q2**: The authors state in L465-466: “Finally and not surprisingly, all types of graph transformers cannot learn even very short dependencies”. Can the authors provide a more detailed insight into why this result is unsurprising? The GPS model, for example, uses a local message-passing module that should at the very least match the performance of the vanilla GatedGCN. I find that this warrants further analysis. One possible reason could be the possibly low amount of data seen during training; see related **Q1**.

---

### Official Review · Reviewer_Je5y · 2024-11-03

**Soundness:** 3
**Presentation:** 3
**Contribution:** 3
**Rating:** 6
**Confidence:** 4

**Summary:**

This paper introduces GLoRa, a synthetic benchmark designed to evaluate the ability of graph neural networks (GNNs) to capture long-range dependencies. By generating controlled graph examples with enforceable dependency lengths, GLoRa addresses a key gap in current GNN benchmarks. The authors provide theoretical guarantees for GLoRa, showing that models must capture dependencies of exact lengths to perform well. The paper also presents an empirical evaluation of several GNN architectures (including vanilla, over-smoothing-mitigated, over-squashing-mitigated, and Transformer-based GNNs) on GLoRa, revealing that none of the models perform well beyond modest dependency lengths.

**Strengths:**

- **Novelty in Benchmark Design**: GLoRa provides a synthetic benchmark with strict, enforceable dependency-length requirements, filling an important gap in current graph benchmarks.
- **Theoretical Guarantees**: The benchmark’s theoretical properties, including enforceable dependency lengths, are rigorously proven, making GLoRa a well-grounded tool for long-range dependency evaluation.
- **Clarity and Structure**: The paper is well-structured, with clear explanations of the benchmark construction process and theoretical foundations.

**Weaknesses:**

1. **Disconnection Between Theory and Experiment**: The experiments do not fully validate the theoretical properties of GLoRa, such as the enforceable dependency lengths. Testing trained models across a range of dependency lengths or with varied “holes” in paths might provide empirical support for the benchmark’s theoretical claims.

2. **Unexpected Performance of Transformer-Based Models**: Transformer-based GNNs perform poorly on GLoRa, even for small dependency lengths (e.g., \(d = 3\)). This contradicts their generally strong performance on other tasks, raising questions about whether GLoRa aligns with their strengths or if implementation details (like positional encodings) limit performance. Further exploration of encoding options or discussing the potential limitations of Transformers on GLoRa would clarify this discrepancy.

3. **Limited Testing of Transferability and Practical Relevance**: GLoRa’s relevance to real-world tasks remains untested, as there are no experiments that transfer GLoRa-trained models to practical benchmarks requiring long-range dependencies. Testing transferability on benchmarks like social networks or molecular graphs would substantiate GLoRa’s practical utility.

4. **Missing Important Evaluations for Implicit GNNs**: While the paper tests many GNN models, most of these GNNs do not claim to capture long-distance dependency. Various types of implicit GNNs have been demonstrated to capture long-distance dependency better, but the paper misses this important category of models. A more comprehensive evaluation on implicit GNNs will be helpful.

While GLoRa is a theoretically grounded and novel benchmark for evaluating long-range dependencies in GNNs, the experimental design could be strengthened to better align with its theoretical properties and validate practical relevance. Specifically, testing models across varying dependency lengths, addressing the Transformer performance anomaly, and exploring GLoRa’s transferability to real-world tasks would greatly enhance the impact of this work.

**Questions:**

1. Could you clarify why Transformer-based models perform poorly on GLoRa, even at small dependency lengths? Have alternative positional encodings or adaptations been considered?
2. How does GLoRa handle variations in the number of “holes” within paths, and would testing different numbers of interruptions provide further insight into model performance?
3. Are there plans to test models that perform well on GLoRa against real-world benchmarks requiring long-range dependencies to validate GLoRa’s practical transferability?
4. How do various types of implicit GNNs perform on the proposed benchmark?

---

### Official Review · Reviewer_jY5f · 2024-11-04

**Soundness:** 3
**Presentation:** 3
**Contribution:** 3
**Rating:** 8
**Confidence:** 3

**Summary:**

The paper presents an algorithm for generating a synthetic dataset for every dependency length and demonstrates how to use this benchmark to identify, with certain guarantees, the maximum dependency length that a graph learning system can learn. Additionally, the paper illustrates the application of the benchmark in the experiment section.

**Strengths:**

The method is novel, and the paper is well-written. The methodology is easy to follow, and the experiment section is well-structured and clearly presented. Through dedicated experiments, the authors show that, in nearly all cases, the performance degradation with increasing dependency length cannot be attributed to any of the three phenomena: over-smoothing, over-squashing, or vanishing gradients. This finding opens up two directions for future research: identifying the true causes of this degradation and developing methods to address it.

**Weaknesses:**

Overall, the paper is good; however, some improvements are needed in figure presentation. For example, the position of subgraph titles should be consistent across Figures 2, 3, and 4.

**Questions:**

See weaknesses.

---

### Public Comment · ~Peter_Bloem1 · 2025-05-19
**Camera ready version still includes proof sketch**

Is it possible that something has gone wrong in uploading the camera-ready? It does not seem to contain the full proof that the meta-review refers to. I would be interested in reading this.

I'm also curious why I cannot see the rebuttals here, when I can see them in other papers, but perhaps this is a setting the authors chose for their rebuttal comments...

---

### Public Comment · ~Kunyang_Zhou1 · 2026-02-04
**Benchmark link on Github is unavailable.**

Dear authors

I find that the benchmark link on github is unavailable. The link is https://github.com/DongzhuoranZhou/GLoRa. Could you share the available link?  Thanks a lot.

Best

---

> ### Public Comment · ~Dongzhuoran_Zhou1 · 2026-02-12
>
> Thank you for pointing this out, and I sincerely apologize for the inconvenience. The benchmark repository is temporarily unavailable and will be restored by the end of February, as I am currently handling several other deadlines. I appreciate your patience.

---

### Meta-Review · Area_Chair_vbnP · 2024-12-23

**Metareview:**

This paper introduces a new benchmark for learning long range dependencies in graphs, a notoriously challenging problem. The reviewers agreed that the paper addresses an important topic, is well written, and makes a number of meaningful contributions.

There were some concerns about the gap between the model in the benchmark and practice, and issues with the rigor of the theory in the paper. Most of these concerns were addressed in the rebuttal and the reviewers increased their scores accordingly. It will be critical to  address all of the reviewer concerns in the final version of the paper, especially the requested changes to Theorem 1 which multiple reviewers pointed out was not rigorous.

**Additional Comments On Reviewer Discussion:**

The authors did a good job of addressing the reviewer concerns, including a new proof of Theorem 1 which was requested by reviewers.

---

### Decision · Program_Chairs · 2025-01-22

Accept (Poster)